

# A Jack of All Trades—Tawaki/Fiordland penguins are able to utilise diverse marine habitats during winter migration

Thomas Mattern[1,2,3], Klemens Pütz[4], Pablo Garcia Borboroglu[2,5], Ursula Ellenberg[2,3,6], David M. Houston[3,7], Bernhard Lüthi[4] and Philip J. Seddon[1]

[1] Department of Zoology, University of Otago, Dunedin, Otago, New Zealand
[2] Global Penguin Society, Puerto Madryn, Chubut, Argentina
[3] The Tawaki Trust, Dunedin, Otago, New Zealand
[4] Antarctic Research Trust, Zürich, Switzerland
[5] Centro Nacional Patagónico (CONICET), Puerto Madryn, Chubut, Argentina
[6] Department of Marine Science, University of Otago, Dunedin, Otago, New Zealand
[7] Department of Conservation, Auckland, New Zealand

Corresponding author
Thomas Mattern,
t.mattern@eudyptes.net

## ABSTRACT

Migration and non-breeding movements are common across animal groups and are often driven by seasonal changes in habitat conditions. This behaviour is prevalent in crested penguins (*Eudyptes* sp.), which have evolved in and still primarily inhabit the subantarctic regions of the Southern Hemisphere. These species migrate outside the reproductive phase due to the limited year-round productivity around the breeding sites. Tawaki/Fiordland penguins (*Eudyptes pachyrhynchus*) are unusual in that they breed in temperate, continental New Zealand, an environment that appears productive enough to support year-round residency, yet they undertake extensive migrations during the non-breeding period. To investigate the drivers and patterns behind this behaviour, we used satellite telemetry to track 14 adult tawaki from across their breeding range during the winter of 2019. We examined whether migration routes differed by breeding location, and used maximum entropy (Maxent) modelling to identify environmental predictors of habitat use during the non-breeding period. All penguins followed a similar south-westerly trajectory toward the subantarctic waters south of Tasmania, irrespective of origin. Birds reached maximum distances of up to 2,193 km from their colonies, traveling a median total distance of 6,086 km over 135 days. Maxent models showed that mixed layer depth (*i.e.*, the mixing height at the ocean surface) around 80 m was the strongest predictor of habitat suitability, aligning with known foraging depths in this species. Tawaki were associated with oceanic habitats ranging from polar to subtropical regions—a broader environmental range than other crested penguins, which tend to remain within a single water mass. These findings highlight the flexibility of tawaki in their use of marine habitats. This behavioural plasticity may suggest resilience to environmental variability, offering insights into why tawaki appear to be maintaining stable population trends while other New Zealand crested penguins are in decline.

## INTRODUCTION

Migration and non-breeding dispersal is common in many animal groups, but is particularly prevalent in birds (*Newton, 2007*). With the ability to fly, birds can cover vast distances resulting in collective travel routes spanning the globe (*Shaffer et al., 2006*; *Quillfeldt, Voigt & Masello, 2010*; *Weimerskirch et al., 2015*). Migratory movements are often apparent in species that live and breed in regions affected by seasonal resource variability, effectively forcing the animals to travel to more productive environments at certain times of the year (*Bowler & Benton, 2005*; *Newton, 2007*; *Ainley & Wilson, 2023*). Because long distances might need to be travelled, the migration phase of a species' annual life-cycle is often energy demanding (*Arizmendi-Mejía et al., 2013*), requiring animals to optimise their migration movements accordingly (*Hennin et al., 2016*). This should be especially relevant for non-flying bird species such as penguins with limited migratory ranges.

Of the 18 species of penguins listed on the IUCN Redlist (iucnredlist.org), six species are considered sedentary, remaining at their breeding sites all year without migratory movements being inherent part of their annual cycle, whereas the remainder disperse when not breeding (*Garcia Borboroglu & Boersma, 2013*). Seasonal migration is generally more prevalent in penguin species that breed at higher latitudes, particularly in or near Antarctica. This pattern is often linked to breeding sites becoming ice-bound in winter (*Trathan & Ballard, 2013*; *Wienecke, Kooyman & LeMaho, 2013*), or to seasonal changes in prey availability (*Pütz et al., 2006*; *Raya Rey, Trathan & Schiavini, 2007*). However, there are notable exceptions to this pattern. Gentoo penguins (*Pygoscelis papua*), for example, that co-exist with crested penguins (*Eudyptes* spp.) in various locations in the subantarctic region or with other *Pygoscelis* species in Antarctica, do not disperse over long distances away from their colonies outside of the breeding season, whereas all Eudyptes undertake obligate migration (*Croxall & Davis, 1999*). Conversely, on the New Zealand mainland, adult hoiho/Yellow-eyed penguins (*Megadyptes antipodes*) and kororā/Little penguins remain at or near their breeding colonies throughout the non-breeding period (*Wilson & Mattern, 2019*; *Hickcox et al., 2022*). Tawaki/Fiordland penguins (*Eudyptes pachyrhynchus*, referred to as "tawaki" hereafter) also breed along the mainland coast but, unlike the other two species, disperse far into the subantarctic region outside the breeding season (*Mattern et al., 2018*; *Thiebot et al., 2020*; *Green et al., 2022*). This highlights that latitudinal and seasonal environmental variability alone cannot account for the migratory behaviour of all penguin species. Tawaki, in particular, deviate from expected patterns warranting closer examination of their non-breeding movements.

After completion of the breeding season in December, at the height of the austral summer, tawaki leave their breeding colonies for ca. 10 weeks in preparation for their annual moult, which the birds generally complete when back in their colonies (*Mattern, 2013*; *Mattern & Wilson, 2019a*). Given the high energetic demands associated with both breeding and moulting (*Brown, 1989*), it is unexpected that tawaki undertake long-distance migrations during the pre-moult period, especially as marine productivity around the New Zealand mainland peaks at this time (*Murphy et al., 2001*; *Goebel, Wing & Boyd, 2005*). Rather than taking advantage of locally abundant resources, the penguins travel southwest into the

subantarctic region and forage in water masses influenced by the Antarctic Circumpolar Current (*Mattern et al., 2018*). This behaviour suggests that seasonal increases in local productivity are not the primary driver of tawaki migration—at least not during the pre-moult phase.

Understanding the drivers behind penguin migration ultimately requires knowledge of the prey resources targeted at non-breeding destinations (*Croxall & Davis, 1999*). However, obtaining such information is logistically challenging, as direct observation and sampling in remote wintering areas are rarely feasible. An alternative is to evaluate habitat use in relation to a species' known foraging ecology. The marine ecology of tawaki during the breeding season has received scientific attention in the past decade, providing detailed insights into their diving behaviour and habitat use. Tawaki exhibit considerable behavioural plasticity at this time, foraging across a broad range of marine environments; from shallow inshore and coastal shelf waters (*Mattern & Wilson, 2019a*), to fjord systems and pelagic zones (*Hornblow, 2022*; *Otis et al., 2025*). This ecological flexibility raises the possibility that individuals from different breeding regions may also adopt varying strategies during the non-breeding period. Yet previous tracking studies of tawaki migration have focused on single sites (*Mattern et al., 2018*; *Thiebot et al., 2020*), potentially limiting our understanding of the full range of their migratory behaviour. In this study, we address this gap by tracking birds from across the species' breeding range and by testing for variation in migration patterns associated with both geographic origin and sex.

We used satellite telemetry to examine the winter movements of tawaki in 2019. Our objectives were to: (a) track individuals from multiple breeding sites across the species' range to assess whether migratory patterns vary with geographic origin; (b) identify key environmental characteristics (including sea surface temperature, bathymetry, and mixed layer depth) associated with their non-breeding destinations using habitat suitability modelling; and (c) to explore whether tawaki's broader habitat use sets them apart from other crested penguins in terms of ecological flexibility.

## MATERIALS & METHODS

### Study species

Tawaki are one of seven crested penguin species (*Eudyptes sp.*) currently recognized by the IUCN Redlist (https://www.iucnredlist.org/) and the only crested penguin that breeds in a temperate and continental setting—that is, on the mainland coast of Aotearoa/New Zealand (NZ), rather than on remote subantarctic islands (*Mattern & Wilson, 2019a*). Long thought to be one of the rarest penguin species, recent population surveys revealed their numbers to be considerably larger than previously assumed with estimates ranging up to 50,000 mature individuals (*IUCN, 2020*). Exact population estimates are difficult due to the species' cryptic breeding behaviour in remote and difficult to access regions of the NZ South Island (*Mattern & Wilson, 2019a*).

Tawaki have been described as "winter breeders" (*Poupart et al., 2019*) although this overly generalises the species' annual cycle with eggs hatching in the early austral spring (September) and chicks fledging in early summer (December). At the conclusion of the

breeding season, the penguins undertake an extensive pre-moult migration that lasts until late January and early February before undergoing their annual moult (*Mattern et al., 2018*). By the end of February and beginning of March, tawaki leave their colonies on their winter migration. Throughout this study, we use the term "winter migration" to refer to the non-breeding migration of tawaki, which begins in the austral autumn (March–May) and extends into winter. While not strictly confined to the winter months, this terminology aligns with common usage in seabird migration literature.

## Satellite transmitter deployments

Between 22 February and 7 March 2019, a total of 16 adult tawaki that had completed their annual moult as indicated by the fresh condition of their plumage were fitted with Argos satellite transmitters (SPOT-275 Platform Transmitter Terminals, dimensions: W × L × H–15 mm × 85 mm × 17 mm; weight: 40 g; Wildlife Computers, Redmond, WA, USA). All birds were captured by hand at the moulting sites, usually their nesting burrows, and then weighed with 5 kg Pesola spring balance (accuracy: 50 g), individually marked with passive integrated transponders, and morphometric measurements taken to determine sex (*White et al., 2021*). The devices were attached using the Tesa-tape method (*Wilson et al., 1997*) with a thin layer of Pattex rubber glue (Henkel AG & Co. KGaA, Düsseldorf, Germany) applied to the feathers below the transmitter base, with an additional layer of Araldite 5-minute epoxy glue (Selleys, Auckland, New Zealand) covering the tape wrapped around the devices. The entire deployment procedure from catching to release took between 10–15 min, during which the birds were kept in a cloth bag that covered their heads. Birds were eventually released back into the burrow in which they had moulted.

Deployments occurred at three sites in the NZ Southwest (Fig. 1). We aimed to deploy five devices at each of three sites spanning the species' breeding range, namely Jackson Head, West coast (S43.9633, E168.6107, "Jackson Head"); Harrison Cove in Milford Sound/Piopiotahi, Fiordland (S 44.6202, E167.9097, "Milford Sound"); and Whenua Hou, Foveaux Strait (S46.7582, E167.6407, "Whenua Hou"). However, Jackson Head was practically devoid of moulting birds when visited so that only a single female could be fitted with a transmitter. Instead, tthree of the remaining devices were deployed on additional birds on Whenua Hou, while another two transmitters were fitted to two tawaki that moulted outside of the tawaki breeding range in the care of the Oamaru Blue Penguin Colony on the east coast of the South Island (S45.1103, E170.9801). The two Oamaru deployments yielded very little data. One bird ceased transmitting shortly after release presumably because it managed to preen off the device; the second bird travelled for 19 days along an south-eastward trajectory and transmission stopped after 19 days (see https://ptx.lat/tas19). The unusual behaviour of this bird triggered a follow-up tracking study to examine whether rehabilitation may alter their migration movements. In the context of this study, however, it was decided to omit both data sets from further analysis. Overall, three female and two male penguins from Milford Sound (22.02.2019), three females and five males from Whenua Hou (25.02.2019), and a single female from Jackson Head (27.02.2019) were tracked with SPOT-tags (Table 1).
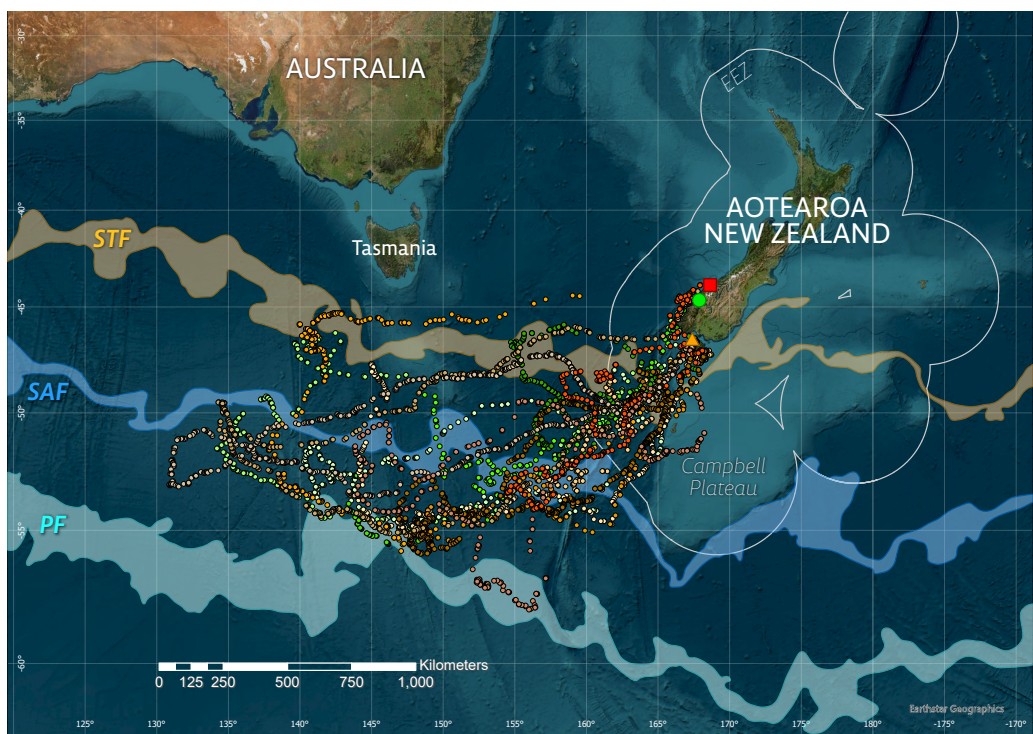

**Figure 1** **Overview of study sites (square–Jackson Head, circle–Milford Sound/Piopiotahi, triangle–Whenua Hou/Codfish Is) and tawaki movements during the 2019 winter dispersal period (March–August).** Only the three tagging sites are shown for clarity, though tawaki breed at additional locations along the Fiordland coast and Stewart Island. Satellite fixes after filtering are shown as points with different hues representing the respective deployment areas within the species' breeding range, *i.e.,* Jackson Head–red ($n = 1$), Milford Sound–green ($n = 5$), Whenua Hou–orange ($n = 8$). The three major oceanic fronts–the Subtropical Front (STF, area between sea surface temperature contours 11–12 °C), Subantarctic Front (SAF, 8–9 °C) and polar front (PF, 3–4 °C)–are indicated as transparent areas. The circular outlines around Aotearoa/New Zealand denote the Exclusive Economic Zone (EEZ).

## Basic satellite data analysis

The satellite transmitters were programmed to start operating once the device's saltwater switch detected immersion, *i.e.,* when a penguin had finished moult and entered the sea to launch the winter journey. The devices were programmed to transmit to Argos satellites up to 15 times per hour, a rate calibrated to balance data acquisition with energy consumption and allow for battery life of up to six months, sufficient to cover the entire winter migration period. However, the actual number of received location fixes was typically lower due to factors such as satellite pass frequency, sea state, and antenna orientation, which can affect signal reception. With these settings an average 10 locations per day could be obtained for each bird. Not all these locations were classified as reliable by the Argos system (*e.g.,* *Thomson et al., 2017*). Therefore, for spatial analysis any locations for which the locational error could either not be determined (Argos classes A, B, Z) or was >1,500 m (Argos class 0) were omitted from analysis. Accepted satellite data was furthermore filtered using the function 'sdafilter' from the package 'argosfilter' (version 0.70, *Freitas, 2022*) in R
4.2.2 (*R Core Team & R Development Core Team , 2022*). This function removes locations identified as outliers based on swimming speed, distance between successive locations, and turning angles, using dynamic thresholds calculated by an algorithm described in *Freitas et al. (2008)*. Filtering was based on a maximum plausible swimming speed of 2 m/s (7.2 km/h), consistent with the default value implemented in 'argosfilter'. After filtering, each tag yielded a median of 14 location fixes per bird per day (range: 3–30 fixes). For eight individuals, filtering resulted in data gaps of 1–6 days (median: 1 day), while the remaining eight birds produced daily location fixes until transmission ceased.

Accepted data were then used to determine individual tracks and to calculate basic trip statistics, providing basic metrics to compare movement of penguins from the different study sites. These were trip duration, maximum distance from start point, distance travelled per day and mean daily swimming speed. These metrics allow comparisons across individuals, even when tracking durations vary or are incomplete. Swimming speed was calculated as the distance between two consecutive locations on the same day divided by the time difference when these two locations were recorded. These speeds must be considered conservative for two reasons: first, because the calculation does not account for deviations from straight-line travel between fixes, and second, because they exclude the additional vertical and lateral movements associated with diving behaviour. We used general linear mixed models, employed in R to examine differences in trip statistics between birds from the different sites.

## Maxent modelling of suitable migration destinations

We used the maximum entropy approach (''Maxent''; *Phillips, Anderson & Schapire, 2006*; *Phillips & Dudík, 2008*; *Phillips et al., 2017*) to model the distribution of tawaki at their migration destination in the subantarctic region. Maxent employs a machine-learning method that uses presence-only data to estimate a target probability distribution by finding the probability distribution of maximum entropy. It examines an a priori set of environmental variables at locations where animal presence has been recorded and compares them to the same variables at randomly generated pseudo-absence locations. This allows identification of variables that best explain habitat selection and enables projection of suitable conditions across broader areas where no presence data exist. Maxent's similarity to inhomogeneous Poisson point processes means that outputs can be scaled to probability of presence using a complementary log–log link (*cloglog*) function so that habitat suitability can be visualised in a 0–1 raster set, where cell values closer to 0 are indicative of unsuitable, and values closer to 1 of suitable habitat conditions.

Six environmental features previously identified as important predictors of tawaki habitat use during their pre-moult migration (*Mattern et al., 2018*) were selected to develop a Maxent distribution model for their winter migration. This allowed us to assess whether environmental preferences inferred from one non-breeding stage also apply during a different phase of the annual cycle. Two features were derived from NASA's satellite-based AQUA-Modis ocean colour program (https://oceancolor.gsfc.nasa.gov), namely night-time sea surface temperature (*nsst*), a proxy for surface water mass characteristics, and chlorophyll-a concentration (*chlo_a*), an indicator of surface

Peer*J*

**Table 1  Overview of individual trip statistics of 14 adult tawaki performing their winter migration between February and August 2019.** Values marked with asterisk indicate values derived from incomplete data due to transmission cessation before trip phase was completed. Rows with bold font indicate data derived from complete trips, *i.e.* trips where the bird returned to the mainland while still transmitting. Note that 'Transmission Duration' equals 'Trip Duration' in complete trips. The final row provides the mean ± standard deviation for each variable across all individuals.

| BirdID | Sex | Body mass (kg) | Trip start (dd mm) | Trip end (dd mm) | Last fix (dd mm) | Transmission Duration (days) | Minimum distance covered (km) | Max Range (km) | Outward (days) travel (days) | At destination (days) | Return travel (days) |
|---|---|---|---|---|---|---|---|---|---|---|---|
| *Jackson Head (North)* | | | | | | | | | | | |
| **Hedwig** | **f** | **2.4** | **28.02.** | **03.08.** | | **156** | **5,012** | **1,480** | **49** | **29** | **78** |
| *Milford Sound (Central)* | | | | | | | | | | | |
| **Jürg** | **m** | **3.6** | **23.02.** | **18.07.** | | **145** | **7,200** | **2,193** | **35** | **46** | **64** |
| Verena | f | 2.5 | 24.02. | – | 07.06. | 103 | 5,315 | 2,585 | 34 | 65 | 4* |
| K-TEC | m | 3.6 | 02.03. | – | 11.05. | 70 | 2,942 | 1,586 | 33 | 37* | – |
| Heidi | f | 3.1 | 05.03. | – | 26.04. | 52 | 2,577 | 2,043 | 34 | 18* | – |
| Andrea | f | 3.2 | 25.02. | – | 20.07. | 145 | 4,603 | 1,116 | 34 | 80 | 31* |
| *Whenua Hou (South)* | | | | | | | | | | | |
| Floyd | f | 2.7 | 28.02. | – | 06.05. | 67 | 3,650 | 1,838 | 34 | 33* | – |
| **Toby** | **m** | **2.6** | **26.02.** | **13.07.** | | **137** | **3,917** | **1,002** | **42** | **35** | **60** |
| Janis | f | 2.8 | 26.02. | – | 10.07. | 134 | 5,985 | 2,217 | 32 | 81 | 21* |
| Sherlock | m | 3.5 | 06.03. | – | 15.05. | 70 | 3,185 | 1,595 | 46 | 19 | 5* |
| **Ueli** | **m** | **3.1** | **04.03.** | **13.07.** | | **131** | **6,894** | **1,689** | **31** | **39** | **61** |
| Wulfers | m | 3.6 | 28.02. | – | 30.06. | 122 | 6,815 | 2,461 | 32 | 42 | 48* |
| Talix | f | 2.7 | 28.02. | – | 01.07. | 123 | 6,498 | 2,688 | 41 | 42 | 40* |
| **Tucnak** | **m** | **3.3** | **03.03.** | **13.07.** | | **132** | **6,086** | **1,412** | **18** | **55** | **59** |
| Mean ± SD | | 3.1 ± 0.4 | 28.02. ± 3d | 18.07. ± 8d | – | 113 ± 34 | 5,049 ± 1,587 | 1,850 ± 530 | 35 ± 7 | 48 ± 19 | 64 ± 7 |
ocean productivity. We also used sea level anomaly (*sla*; Copernicus Climate Data, https://doi.org/10.24381/cds.4c328c78) and surface current velocity (*velo*; OSCAR 3rd degree surface currents, https://podaac.jpl.nasa.gov/dataset/OSCAR_L4_OC_third-deg), which both reflect mesoscale oceanographic features such as eddies. Mixed layer depth (*mld*; SEANOE, https://doi.org/10.17882/91774) represents vertical structure and nutrient mixing potential, and bathymetry (*bathy*; GEBCO; https://www.gebco.net/data_and_products/gridded_bathymetry_data/), capturing seafloor topography and proximity to continental or shelf edges.

Tawaki disperse south over the austral fall and winter period, which greatly limits completeness of satellite-based environmental data. Short day lengths and frequent cloud cover means that raster layers that are derived from optical measurements from space often contain cells without valid data and, thus, are not suitable for Maxent modelling. To overcome this limitation, we had to use seasonal averages that combine measurements taken from March to June 2019. *nsst* and *chlo_a* data were directly available for download as seasonal averages for the focal period. However, current velocity and sea level anomaly data were only downloadable as 5-day grids. For these variables, all data sets for March to June were downloaded and combined into a single mean raster by using the 'Raster Calculator' in ArcGIS Pro 3.1.0 (Esri Inc., Redlands, CA, USA). Sea surface current data consisted of vertical (v) and horizontal component (u) rasters. Using the 'Raster Calculator' the vectorial current velocity raster was calculated as velocity $= \sqrt{(u^2 + v^2)}$. The mixed layer depth raster was only available as a composite of direct measurements taken between 1970 and 2021 so that seasonal discrimination of the data was not possible. Obviously, bathymetry also is a temporally static variable.

A limitation of our modelling approach is that environmental variables were available only as seasonal composites, due to persistent cloud cover and limited daylight during the study period. As a result, we were unable to assess short-term environmental variability or dynamic responses to transient oceanographic features, such as eddies or frontal shifts. This constrains our ability to interpret fine-scale drivers of habitat selection and may obscure individual differences in response to short-lived features. However, the use of averaged data still permits adequate identification of general habitat preferences at the regional scale.

All environmental rasters were clipped to the same extent of 30°S to 65°S and 80°E to 45°W (*i.e.*, across the date line) in ArcGIS (function 'Clip raster'). As the data sets were available in different spatial resolutions, the respective rasters were first projected to UTM datum using the ArcGIS function 'Project raster', then resampled ('Resample') to have the same resolution with cell sizes of 25 × 25 km. The resulting clipped and resampled rasters were then exported as ASCII raster files that could be processed by the Java software Maxent v3.4.4 (*Phillips, Dudík & Schapire, 2023*).

Tawaki migration movements can be broadly differentiated into three phases: an outbound travel phase, a mid-trip phase in the distant subantarctic region, and a return phase to the colony (*Mattern et al., 2018*). Given the temporal limitations of Argos data and the absence of high-resolution behavioural metrics that would be required for state-based definitions, we classified locations as "at destination" if they fell beyond 75% of the bird's maximum distance from its point of origin. This threshold-based approach

provides a consistent operational definition for identifying the central portion of each track, although foraging almost certainly occurs throughout all phases of the migration. Only locations classified as "at destination" were used to model environmental suitability in the subantarctic region.

To reduce temporal autocorrelation in the movement data, we calculated one average location per day for each individual. This subsampling approach reduces serial dependence between consecutive Argos fixes which are otherwise closely spaced in both time and space. While some spatial and inter-individual dependence may remain, this level of thinning is commonly used to reduce pseudo-replication in presence-only models derived from telemetry data (*Elith et al., 2011*; *Thurfjell, Ciuti & Boyce, 2014*). The resulting dataset balances the need for spatial independence with adequate coverage of the birds' distribution at their non-breeding destination.

The Maxent analysis was set up to use a randomly chosen 25% of the location data as test samples with the remaining data used for model training over 500 iterations. The model produces response curves for each environmental variable that are then used for jack-knife analysis of variable importance expressed as percent contribution to the model (*Phillips, 2017*).

### Permits and animal ethics

This study complies with the relevant national, international (*Field et al., 2019*), and institutional guidelines regarding animal care. It was conducted under a research permit (38882-RES) issued by the Department of Conservation under the New Zealand Wildlife Act 1953. All manipulations were approved by the Ethics Committee of the University of Otago (D69/17).

## RESULTS

### Satellite tracking

Between 23 February and 11 September 2019, a total of 15,010 locations was received for 14 tawaki fitted with satellite tags. After filtering, 5,024 locations (retention rate: 33.5%) were accepted for the reconstruction of the birds' travel paths (Fig. 1) and subsequent analysis.

Birds from Milford Sound/Fiordland ($n = 5$) and Whenua Hou/Foveaux Strait ($n = 8$) departed on their winter migration at similar times, with average departure dates of 27 February and 1 March, respectively. One bird from Jackson Head on the West Coast ($n = 1$) departed on 28 February; while this individual was included for completeness, no group-level inference was drawn from this site.

A median of 279 locations (range: 98–870) were received per bird and penguins transmitted for a median of 127 days (range: 52–192 days) which allowed the reconstruction of complete winter trips for five birds, with an additional six birds transmitting well into the return stage of their journey. Three birds stopped transmitting before they had embarked on their return to the mainland.

All tawaki travelled southwest with destinations located in the subantarctic region approximately 1,000 km due south of Tasmania in international waters outside of the NZ Exclusive Economic Zone (Fig. 1). Penguins for which complete winter trips could be

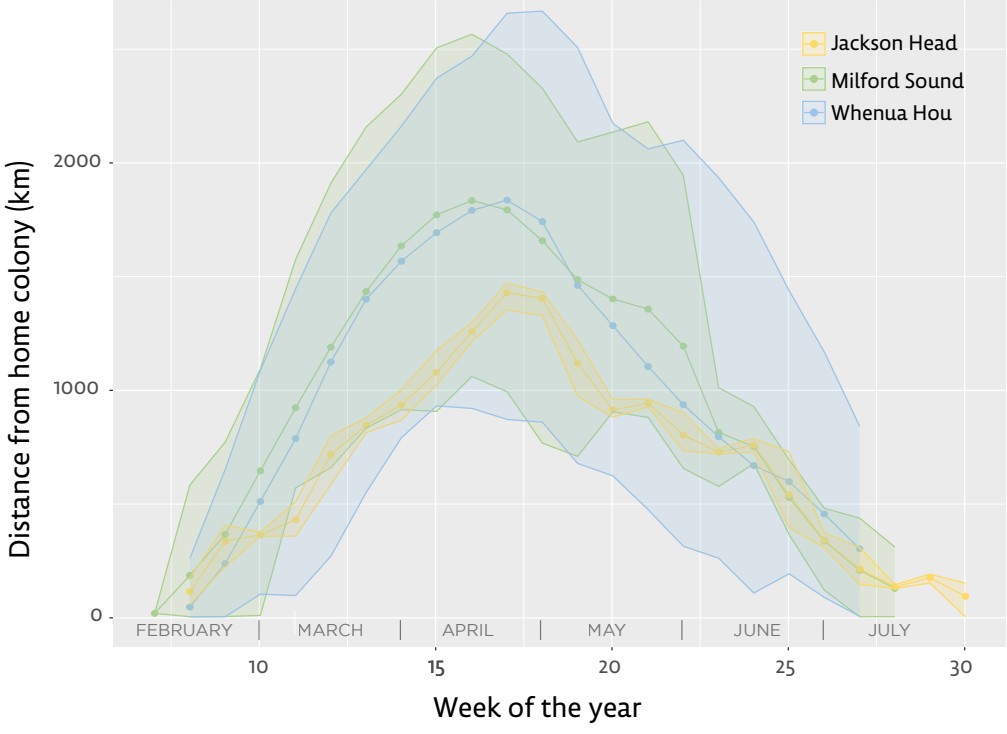

**Figure 2** **Distance from home colony for the three main groups of tawaki tracked with satellite transmitters during their winter migration 2019 (March–August).** Lines indicate the mean distance from the home colony calculated for each group during distinct weeks of the year (green–Milford Sound, $n = 5$; blue–Whenua Hou, $n = 8$; yellow–Jackson Head, $n = 1$). Shaded areas represent the range between mean minimum and maximum distances recorded during the respective weeks of the year. Note that Jackson Head data derived from a single bird. Jackson Head and Milford Sound are located 320 km and 240 km, respectively, north of Whenua Hou.

recorded achieved maximum distances of between 1,002 and 2,193 km (median: 1,585 km, $n = 5$) away from their point of origin, covering total distances of a median 6,086 km (range: 3,917–7,200 km) over the course of 131–156 days (median: 135 days) (Table 1). Penguins that entered the return phase before transmissions stopped reached a median maximum distance of 1,689 km (range: 1,002–2,688 km, $n = 11$) from their place of moult.

Comparing the birds' mean distances from their place of moult on a weekly basis shows little difference in the distances reached each week when comparing birds from Milford Sound and Whenua Hou (Fig. 2). Penguins from Milford Sound and Whenua Hou had reached their non-breeding destinations by the first week of April. The return journeys in both groups started as early as the first week of May, with all birds moving back towards the mainland by the first week of June. The female from Jackson Head did not distance herself as much or as fast from the mainland, but her distances from the colony still fall well within the range of the other groups.

Linear mixed effect models of the birds' spatial distribution using the means of recorded longitude and latitude show no major differences in the distribution of penguins from the three main groups during any of the trip stages (Table 2). The only significant difference

**Table 2** Linear mixed-effects models of the spatial distribution of 14 tawaki fitted with satellite transmitters during their winter dispersal (February–August 2019). Satellite data was split into the three main stages of the winter dispersal, *i.e.*, outgoing ($n = 14$ birds), at dispersal destination ($n = 11$), and return ($n = 5$); only data representing the entire stage of dispersal were used (see also Table 1). The model uses group (Jackson Head, Milford Sound, Whenua Hou) and sex as fixed effects, and BirdID as random effect. Note that sex had to be removed from the models for the Returning stage as the data limitation resulted in a confounding effect of the variable (only one female left in data set).

| PARAM ~GROUP+SEX+(1\|BIRDID) | | | | | |
|---|---|---|---|---|---|
| | Estimate | Std Error | DF | t | p |
| *OUTGOING* | | | | | |
| Latitude (°S) | | | | | |
| Intercept | −49.29 | 0.73 | 66 | −67.36 | <0.001 |
| Group (Jackson Head) | 0.64 | 1.46 | 10 | 0.44 | 0.669 |
| Group (Whenua Hou) | −1.60 | 0.83 | 10 | −1.93 | 0.082 |
| Sex (Male) | −0.29 | 0.81 | 10 | −0.36 | 0.728 |
| Longitude (°E) | | | | | |
| Intercept | 161.19 | 1.64 | 66 | 98.33 | <0.001 |
| Group (Jackson Head) | 3.65 | 3.32 | 10 | 1.10 | 0.298 |
| Group (Whenua Hou) | −1.58 | 1.86 | 10 | −0.84 | 0.418 |
| Sex (Male) | 1.72 | 1.82 | 10 | 0.94 | 0.368 |
| *AT DESTINATION* | | | | | |
| Latitude (°S) | | | | | |
| Intercept | −49.76 | 0.71 | 69 | −69.95 | <0.001 |
| Group (Jackson Head) | −3.17 | 1.59 | 7 | −1.99 | 0.086 |
| Group (Whenua Hou) | −0.57 | 0.88 | 7 | −0.65 | 0.536 |
| Sex (Male) | −3.09 | 0.84 | 7 | −3.69 | 0.008 |
| Longitude (°E) | | | | | |
| Intercept | 145.14 | 5.35 | 69 | 27.14 | <0.001 |
| Group (Jackson Head) | 11.91 | 10.19 | 7 | 1.17 | 0.281 |
| Group (Whenua Hou) | −2.89 | 6.36 | 7 | −0.45 | 0.663 |
| Sex (Male) | 5.99 | 5.95 | 7 | 1.01 | 0.348 |
| *RETURNING* | | | | | |
| Latitude (°S) | | | | | |
| Intercept | −48.17 | 0.65 | 41 | −73.51 | <0.001 |
| Group (Jackson Head) | 1.20 | 0.89 | 2 | 1.35 | 0.310 |
| Group (Whenua Hou) | −1.48 | 0.76 | 2 | −1.95 | 0.191 |
| Longitude (°E) | | | | | |
| Intercept | 163.11 | 1.56 | 41 | 141.11 | <0.001 |
| Group (Jackson Head) | 0.52 | 1.56 | 2 | 0.34 | 0.769 |
| Group (Whenua Hou) | 0.95 | 1.34 | 2 | 0.71 | 0.554 |

is that males tend to forage around 3 degrees further south while at the winter destination. Thus, data from all groups were pooled to model habitat suitability of the subantarctic ocean region as a non-breeding destination for tawaki.

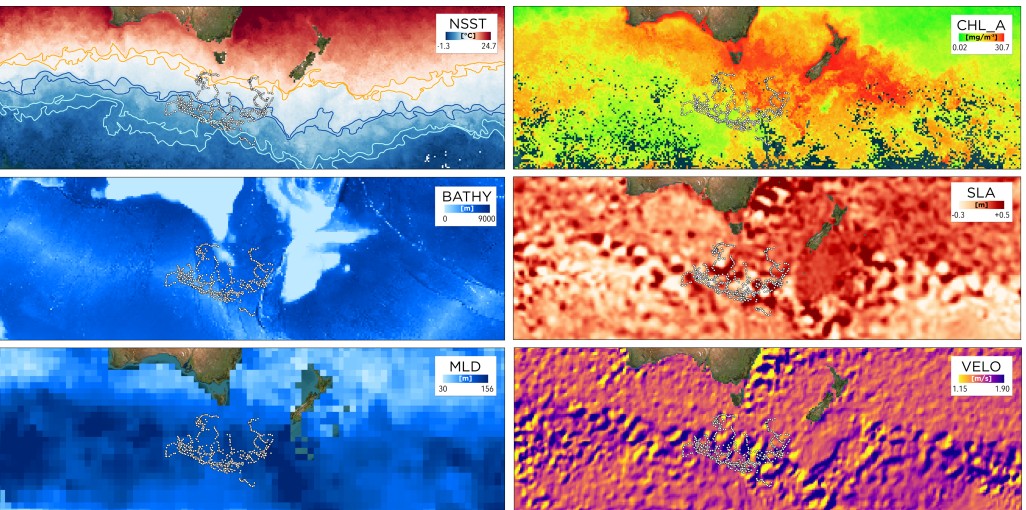

**Figure 3** **Environmental parameters in the southwestern Pacific during March to June 2019 (seasonal average) in relation to tawaki positions determined while the birds were at their dispersal destinations.** White points in each graph indicate daily averaged penguin locations at the dispersal destination used for modelling habitat suitability. The parameters are night-time sea surface temperature (*nsst* (°C)), sea level anomaly (*sla* (m)), bathymetry (*bathy* (m)), mixed layer depth (*mld* (m)), surface current velocity (*velo* (m/s)), and chlorophyll-a concentration (*chlo_a* (mg/m³)). Polygon outlines in the *nsst* graph indicate location of the Subtropical Front (*sst*: 11–12 °C, orange), Subantarctic Front (7–8 °C, blue), and Polar Front (3–4, light blue).

## Maxent modelling of habitat suitability

Calculating daily average locations for each bird and limiting the resulting data set to locations ≥75% of the maximum distance from each bird's point of origin, reduced the data set to 605 locations that were the used to estimate penguin distribution at their winter destination using six environmental variable rasters (Fig. 3).

The Maxent model showed strong predictive performance, with a training AUC of 0.967 and a test AUC of 0.966 ± 0.003, indicating excellent discrimination between suitable and unsuitable habitat. The regularized training gain was 2.282, suggesting a high level of model fit. A total of 325 presence points were used for training and 108 for testing, with 10,256 background points. These results support the robustness of the habitat suitability model and are consistent with standard practices for model evaluation in ecological niche modelling (*Elith et al., 2006*; *Elith et al., 2011*) and comparable to performance reported in other polar marine predator studies (*Ballard et al., 2012*).

The Maxent model (Fig. 4) indicates that suitable habitat for tawaki is located south of the Subantarctic Front, principally south of latitude 50°S and west of longitude 160°E. Overall, bands of suitable conditions can be found along the southern edge of the Subantarctic Front, although their spatial extent and continuity vary across the western Pacific and eastern Indian Ocean. The exception being the area of the Campbell Plateau directly south of New Zealand. Except for a small band of suitable conditions along the south-eastern limits of the Plateau, the subantarctic waters within the NZ EEZ seem largely unsuitable for tawaki.
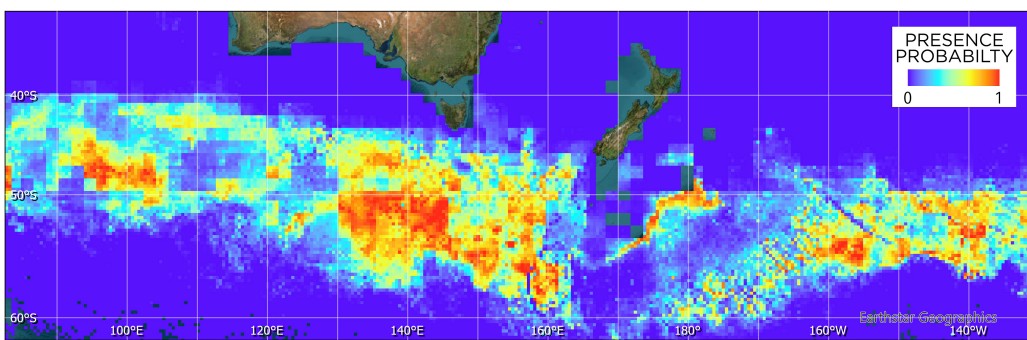

**Figure 4** **Maxent model of tawaki habitat suitability at the penguins' winter dispersal destination based on six different environmental variables (see Fig. 3).** Blue colours indicate zero or close to zero likelihood of tawaki presence, while colours trending towards red denote increasingly suitable habitat and, thus, likelihood of tawaki presence.

The model indicated that mixed layer depth was the most important predictor for suitability of the subantarctic ocean (percent contribution to the model–*mld*: 57%) followed by ocean temperature (*nsst*: 28.3%) and bathymetry (*bathy*: 12.5%). Chlorophyll-a (*chlo_a*: 0.9%), sea level anomaly (*sla*: 0.7%), and surface current velocity (*velo*: 0.6%) played next to no role in determining environmental conditions preferred by tawaki at their winter migration destination. The response curves resulting from the model provide deeper insights into the range over which the environmental variables are of importance. The response curve for mixed layer depth peaks at depths of 81–83 m; most suitable conditions for tawaki presence (probability > 0.75) range at *mld*s between 71 and 91 m (Fig. 5). Sea surface temperature (*nsst*) peaks around 6 °C with probability of presence being higher than 0.75 in water temperatures between 4 °C and 8 °C. For the last relevant variable, bathymetry (*bathy*), has no distinct peak but water depths between 1,800 m and 4,500 m best predict tawaki presence.

## DISCUSSION

Tawaki migrating over winter exhibited similar movement patterns and ranges as has previously been determined for their pre-moult journeys (*Mattern et al., 2018*). However, whereas before the moult the penguins have only 8–10 weeks to complete their journeys, they can take twice as much time over winter. This underlines the remarkability of the tawaki long-distance pre-moult movements (*Mattern et al., 2018*) as well as the importance of the subantarctic region south of Australia for the species' non-breeding distribution and survival (*Thiebot et al., 2020*; *Green et al., 2022*).

### Effects of devices on penguin performance and survival

While externally attached devices inevitably influence the performance of diving animals (*e.g.*, *Chiaradia et al., 2005*; *Wilson & McMahon, 2006*; *Ludynia et al., 2012*), previous studies on crested penguins found little evidence that the effects are significant enough to alter their migratory behaviour or affect their survival (*e.g.*, *Pütz et al., 2006*; *Mattern et al.,*

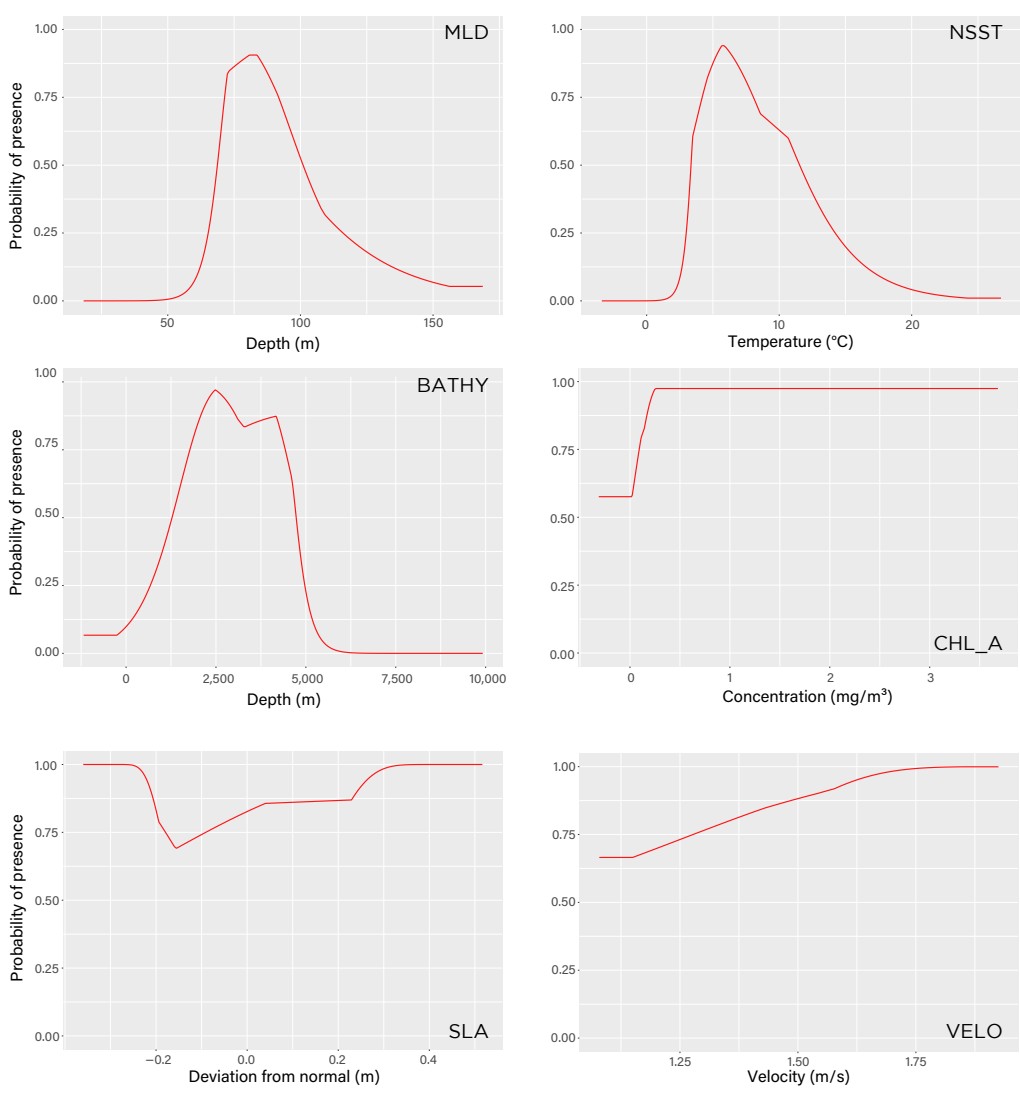

**Figure 5    Maxent model response curves of the six environmental variables used to estimate tawaki presence at winter migration destination.** The parameters are mixed layer depth (*mld*), night-time sea surface temperature (*nsst*), bathymetry (*bathy*), chlorophyll-a concentration (*chlo_a*), sea level anomaly (*sla*), and surface current velocity (*velo*). The curves show the predicted probability of tawaki presence as each variable is varied while keeping the other variables at their average sample value. Graphs are listed in order (left-to-right, top-to-bottom) of contribution importance to the Maxent model (see Results for details).

*2018*; *Houstin et al., 2022*). Although cessation of transmission can indicate the death of a study animal, malfunctions and especially loss of the device are also possible causes (*Sergio et al., 2019*). Devices on five of the eight tawaki from Whenua Hou stopped transmitting before the birds had completed their journeys (Table 1), yet all birds fitted with transmitters in February 2019 were re-sighted in their home colony in August 2019. The single female from Jackson Head was encountered incubating a fresh clutch of eggs in mid-August 2019, with the device still attached and recoverable. Four of the five birds fitted with transmitters

in Milford Sound have subsequently been observed by an automatic wildlife monitoring system, so that only one of the birds fitted with transmitters remains unaccounted for. Hence, device loss was the main explanation for the cessation of signal transmission in this study.

## Movements of penguins from different breeding sites

Tawaki from the species' entire breeding range exhibited similar movement patterns. Unlike patterns observed during the pre-moult migration (*Mattern et al., 2018*), where a dichotomy in migratory strategy was observed depending on breeding status, all birds in this study initiated migration after completing the moult and travelled south of the Subantarctic Front. Although some individuals later relocated toward the Subtropical Front, none foraged exclusively in that zone. This suggests that life-history stage plays a key role in structuring migratory trajectories and destination.

With all birds heading towards the same subantarctic area south of Tasmania (Fig. 1), one might expect penguins from Milford Sound—located approximately 200 km to the north of Whenua Hou—to have travelled farther from their colony on average. However, this pattern is not evident in the weekly distance profiles (Fig. 2), and statistical analysis confirmed that colony origin had no significant effect on displacement during any stage of the migration (Table 2). While group-level means may smooth over individual variability, some Whenua Hou birds did travel farther west than their Milford Sound conspecifics. Nonetheless, such variation appears to reflect individual strategies rather than rather than be a function of those individuals' origins. On the scale of the tawaki winter migration, colony location did not influence overall migratory range.

Recent analyses of genetic diversity across the tawaki breeding range indicate that the species forms a panmictic population, with no detectable genetic differentiation among colonies (*White et al., in press*). This genetic homogeneity suggests a high degree of connectivity among colonies and may help explain the consistent overwinter movement patterns observed across individuals from different breeding sites.

The tawaki winter migration has previously been studied *via* geolocators (*Thiebot et al., 2020*; *Green et al., 2022*). Although not as spatially precise as satellite transmitters, the penguins' reconstructed movement patterns match our observations. While both studies were conducted with birds from colonies in close proximity to or at the same sites of this study, it confirms consistency in the penguins' winter movements. This relative uniformity in the movement patterns of tawaki that moulted along the western shores of southern NZ underlines the importance of the birds' non-breeding destination, the ocean south of the Subantarctic Front.

## Characterisation of environment in the destination region

Physical ocean boundaries, such as fronts, are of significant biological relevance as associated physical processes can lead to nutrient accumulation and increased prey abundance for oceanic predators (*Bost et al., 2009a*). Tracking data showed that the penguins' migration destination was located south of the Subantarctic Front (SAF, Fig. 1), which matches results previously reported using geolocation data (*Thiebot et al., 2020*; *Green et al., 2022*).

This ocean region is characterised by colder surface temperatures (3–8 °C) compared to subtropical waters north of the front, and by relatively low chlorophyll-a concentrations (Fig. 3).

During their winter migration, tawaki leave the areas of subtropical waters north of the Subtropical Front (STF) characterized by temperatures >12 °C, with which the species is exclusively associated during the breeding season (*Mattern & Wilson, 2019a*; *Poupart et al., 2019*; *Hornblow, 2022*; *Otis et al., 2025*). Within 3–4 weeks after their departure most of the penguins passed through the subantarctic region (8–11 °C) and crossed the Subantarctic Front into the waters of the Polar Frontal Zone (PFZ), that is, the oceanic region located between the Subantarctic Front and the Antarctic Polar Front (APF). This is consistent with temperature profiles reported in *Thiebot et al. (2020)*. The PFZ is characterized by an entrainment of nutrients that originate from upwelling forces in the Southern Ocean which can sustain intense diatom blooms (*Sarmiento et al., 2004*). As such, this nutrient richness probably explains why the PFZ is targeted by tawaki.

The closely-related Snares penguins (*E. robustus*), like tawaki, migrate westwards to the ocean south of Australia during winter, but remain primarily along the Subtropical Front (*Green et al., 2022*). However, Snares penguins begin their winter migration 4–6 weeks later than tawaki, reaching their non-breeding destinations only when tawaki are already well into the return phase of their movements. These differences in timing contribute to the spatial segregation between the two species in the eastern Indian Ocean (*Green et al., 2022*). The earlier departure of tawaki in autumn likely allows them to exploit more southerly foraging zones at the PFZ, where longer daylengths still fuel productivity. These conditions may no longer be viable when Snares penguins begin their migration. Shorter daylengths farther south in winter likely also explain the tendency of tawaki to shift toward the STF as the season progresses.

Although chlorophyll-a concentrations are considered a good proxy for ocean productivity and increased prey availability for seabirds (*Suryan, Santora & Sydeman, 2012*), this may not always be the case (*Grémillet et al., 2008*). Moreover, chlorophyll-a concentration data are generally derived from optical satellite measurements and therefore limited to the surface layer of the oceans (*Morales et al., 2011*). As such, chlorophyll-a concentration represents the environmental conditions at the surface and might not reflect what is happening at greater depth. This is likely to be particularly relevant in deep diving species (*Grémillet et al., 2008*), such as tawaki that especially in pelagic environments can dive to depths of >100 m (*Hornblow, 2022*; *Otis et al., 2025*). Although the satellite transmitters did not record diving behaviour, the Maxent model nevertheless provides insight into likely foraging behaviour of tawaki in the subantarctic region.

The models of habitat suitability indicate a substantial effect of the mixed layer depth (MLD) on the likelihood of tawaki presence. With a contribution of 57% to the model, MLD must be described as a stand-out parameter during this second stage of the winter migration. MLD provides an indication of at which depth the thermocline, an abrupt change in water temperature and/or salinity, is located (*Kara, Rochford & Hurlburt, 2000*). Just like oceanic fronts, the thermocline represents a physical boundary at which nutrients and biomass can accumulate (*Bost et al., 2009a*). The model suggests that the highest

likelihood of tawaki being present was in regions with the shallowest MLD over most of the Pacific and Indian Ocean's subantarctic regions (*Sarmiento et al., 2004*). In this region, MLD ranges around 80 m (Fig. 5), which corresponds to dive depths recorded in tawaki during the breeding season when foraging outside of fjord environments (*Hornblow, 2022*; *Otis et al., 2025*). Thus, the thermocline is certainly accessible by tawaki at their migration destination. Foraging at the thermocline has already been described as a strategy used by penguins to pursue predictably-distributed prey (*Bost et al., 2009a*; *Labrousse et al., 2019*). During winter, juvenile emperor penguins are believed to be foraging at the thermocline for myctophid fish and squid (*Labrousse et al., 2019*). Given that the abundance of krill—known to be an important food source for Snares penguin (*Mattern et al., 2009*)—is decreased during winter (*Young et al., 1993*) it seems likely that tawaki are also primarily targeting fish and squid when venturing south. This dietary preference also matches what is known about the species' prey composition during the breeding season (*Van Heezik, 1989*; *Van Heezik, 1990*; *Poupart et al., 2019*; *Hornblow, 2022*).

While MLD emerged as the strongest predictor of habitat suitability, it was modelled using seasonal means, which do not capture short-term variability in thermocline structure. Studies such as *Ballard et al. (2019)* have shown that fine-scale oceanographic features can strongly influence foraging success in penguins. Similarly, the fixed frontal zones depicted in Fig. 1 may not reflect real-time positions of dynamic features such as the Subantarctic or Polar Fronts, which can shift seasonally (*Behrens et al., 2021*). Some distal track boundaries observed in our data may reflect behavioural responses to such dynamic oceanographic constraints. While Lagrangian analyses and high-resolution behavioural data would offer deeper insights into predator–environment interactions (*e.g.*, *Tew Kai et al., 2009*; *Veatch et al., 2025*), obtaining such data in tandem is currently logistically and technologically challenging, particularly in remote oceanic systems like the subantarctic. Nonetheless, these approaches highlight the direction future work might take as data resolution and accessibility improve.

Notably, the Maxent model predicted suitable habitat east of New Zealand, yet none of the tracked penguins visited this region. This highlights the distinction between environmental suitability and realized habitat use. The absence of observed movement into this zone may reflect behavioural conservatism (see also *Mattern et al., 2018*), energetic efficiency in prevailing current systems, or unmodelled ecological constraints. However, tawaki are known to moult along the southeast coast of the South Island (*Mattern & Wilson, 2019a*) which raises the possibility that those birds may utilise these eastern areas.

## Variable ocean habitats, variable diet

Crested penguins all move far from their breeding areas over winter (*e.g.*, *Pütz et al., 2002*; *Rey et al., 2007*; *Bost et al., 2009b*; *Thiebot et al., 2011*; *Green et al., 2022*). Populations that live and breed on subantarctic islands principally show a lateral movement where the birds travel eastwards or westwards, focussing their activities on water masses located at or near frontal zones, be it the Subtropical Front (*e.g.*, Snares penguins, Northern Rockhopper penguins *E. moseleyi*), the Subantarctic Front (Eastern Rockhopper *E. filholi*), and/or the Polar Frontal Zone (Macaroni/Royal penguins *E. chrysolophus/schlegeli*)

(*Green et al., 2023*). What sets tawaki apart from the other crested penguin species, is that their migration spans three major fronts and the associated water masses ranging from subtropical to polar. Foraging in oceanic regions characterised by significantly different environmental conditions should also reflect in the prey consumed by tawaki during their winter migration.

This raises the question, whether prey abundance or quality can explain tawaki travelling thousands of kilometres to the regions south of the Subantarctic Front. Clearly, a substantial amount of prey must be consumed on the penguins' return journey, which, compared to the outgoing phase of the winter migration, is prolonged (Fig. 2). Even though they breed south of tawaki, Snares penguins move further north and remain in the vicinity of the Subtropical Front over winter (*Thompson, 2016*; *Green et al., 2022*). The tawaki satellite tracks show that several of the birds return to the mainland *via* routes along the STF also (Fig. 1), indicating this area is as suitable for tawaki as it is for Snares penguins. Tawaki start on their winter journeys 4–6 weeks earlier than Snares penguins (*Green et al., 2022*) which might make visiting the southern regions in autumn (March–May) more viable for tawaki, as the reduction in ocean productivity during winter has yet to take effect (*Moore & Abbott, 2000*; *Murphy et al., 2001*). However, considering that many other crested penguins forage exclusively at these latitudes through the winter (*Green et al., 2023*), seasonality is unlikely to inhibit habitat use in tawaki and does not convincingly explain their preference for subantarctic over subtropical waters.

Without knowledge of prey consumed during the non-breeding period, it is difficult to determine what drives tawaki to cross two major ocean fronts during their winter migration. However, this extensive use of diverse oceanographic zones may reflect a degree of ecological flexibility that contributes to the comparatively stable status of tawaki relative to other NZ crested penguins.

## Accessing different water masses a key for population stability?

Although tawaki have long been considered one of the rarest penguin species (*McLean et al., 1997*) and one that may be undergoing a steady decline in population numbers (*Otley et al., 2018*), recent population surveys indicate the species is considerably more numerous than previously thought (*Long, 2017*; *Mattern & Long, 2017*; *Long & Litchwark, 2021*), and might even be expanding its range (*Young, Pullar & McKinlay, 2015*; *Mattern & Wilson, 2019a*). As a result, the IUCN Red list downlisted tawaki from "Vulnerable" to "Near Threatened" in 2020 (*IUCN, 2020*). This stands in stark contrast to two other crested penguins breeding in the New Zealand subantarctic region, the Erect-crested penguin (*E. sclateri*) and the Eastern Rockhopper penguin. Both species have experienced significant declines in the past 50 years (*Taylor, 2000*; *Hiscock & Chilvers, 2014*; *Davis et al., 2022*), a trend that largely continues, albeit at a reduced rate in recent years (*Morrison et al., 2015*). They breed on remote subantarctic Bounty, Antipodes, and Campbell Islands southeast of NZ, about halfway between the Subtropical and Subantarctic Fronts. The winter migration of Eastern Rockhopper has been recently examined and birds tended to move eastwards along the Subantarctic Front into the southern Pacific ocean, which conforms with the lateral movement patterns common in crested penguins (*Thompson, 2016*; *Green et al., 2023*).

Data from GLS tracking of Erect-crested penguins seem to suggest similar trajectories in Erect-crested penguins (*Green, 2023*). Hence, the penguins primarily remain within the same water mass throughout the non-breeding period, which also means that changes to the productivity within these water masses may affect their foraging success and survival, ultimately driving population changes (*Hilton et al., 2006*). Environmental phenomena, such as El Niño or La Niña which significantly influence intensity and distribution of ocean productivity, have a more uniform effect within certain water masses (*Racault et al., 2012*), which then in turn could negatively affect penguin species that are concentrating their winter migration within a limited band of latitudes. Compared to other crested penguins found in NZ, tawaki occupy a more diverse array of water masses. It is possible that the behavioural plasticity observed during the breeding season (*Hornblow, 2022*; *Otis et al., 2025*) also supports flexible responses to foraging conditions encountered during the winter migration, although the ecological implications of this remain to be fully understood.

## CONCLUSION

During the non-breeding period, tawaki exhibit diverse movement patterns that suggest a degree of behavioural plasticity in habitat use, similar to what has been observed during the breeding period (*Mattern & Wilson, 2019a*). Such ecological flexibility may represent an advantage under changing ocean conditions, particularly when compared to more specialised strategies observed in other crested penguins in the New Zealand region. According to recent population estimates, tawaki and Snares penguins, two species that breed in the warm waters north of the Subtropical Front, both show stable if not increasing population trends (*Mattern & Wilson, 2019a*; *Mattern & Wilson, 2019b*). Snares penguins even move along the Subtropical Front when not breeding (*Green et al., 2022*). In contrast, Erect-crested and Eastern Rockhopper penguins that both breed exclusively in subantarctic waters are declining (*Hiscock & Chilvers, 2014*). As such, the secret of success seems to lie in access to warmer waters. In this light, tawaki's affinity to move into the subantarctic region appears to be counter intuitive. However, the available data suggest that travelling to the Polar Front does not negatively affect the penguins' survival. This may reflect a combination of favourable physical conditions at the penguins' non-breeding destination and the opportunity to exploit additional resources in warmer waters during their gradual return to the breeding sites. In the end, tawaki's southward journeys during the non-breeding season tie them to the Subantarctic waters—the evolutionary birthplace of crested penguins (*Cole et al., 2019*).

## ACKNOWLEDGEMENTS

We thank the members of our field teams for the assistance with satellite transmitter deployments and recoveries, especially Richard Seed, Andrea Faris, and Mel Young. We are ever so grateful for the logistical support we keep receiving from Southern Discoveries in Milford Sound, and the Department of Conservation crew in Invercargill (Rhuaridh Hannan, Sharon Trainor) that have helped us to get to Whenua Hou for this study. Special

thanks to David Ainley, Katrin Ludynia, Kees H. T. Schreven and two anonymous reviewers for constructive critique and comments of an earlier draft of this manuscript.

### Funding

This study was supported by Erika Bodmer, Sylviane Brunner-Bodmer, Ernst Frei, K-Tec AG, Verena & Ulrich Knoblauch, Alexandra Logan, Jrma Pittard, Eva & Erwin Mayer Anouch & Karin Reckinger, Royal Zoological Society of Antwerp, Klaus Peter Schmidt, Pia & Peter Schudel, Katharina Steiner, Anita & Georg Weibel, Ursula Wölfli and Barbara and Wilfried Wulfers. Field work costs were covered by the Global Penguin Society and supporters of the Tawaki Project on Patreon (https://patreon.com/TawakiProject). The funders had no role in study design, data collection and analysis, decision to publish, or preparation of the manuscript.

### Grant Disclosures

The following grant information was disclosed by the authors:
Erika Bodmer, Sylviane Brunner-Bodmer, Ernst Frei, Verena & Ulrich Knoblauch, Alexandra Logan, Jrma Pittard, Eva & Erwin Mayer, Anouch & Karin Reckinger, Klaus Peter Schmidt, Pia & Peter Schudel, Katharina Steiner, Anita & Georg Weibel, Ursula Wölfli, Barbara and Wilifred Wulfers.
K-Tec AG.
Royal Zoological Society of Antwerp.
Global Penguin Society.
Tawaki Project on Patreon.

### Competing Interests

The authors declare there are no competing interests.

### Author Contributions

- Thomas Mattern conceived and designed the experiments, performed the experiments, analyzed the data, prepared figures and/or tables, authored or reviewed drafts of the article, and approved the final draft.
- Klemens Pütz conceived and designed the experiments, performed the experiments, authored or reviewed drafts of the article, and approved the final draft.
- Pablo Garcia Borboroglu conceived and designed the experiments, authored or reviewed drafts of the article, and approved the final draft.
- Ursula Ellenberg conceived and designed the experiments, performed the experiments, authored or reviewed drafts of the article, and approved the final draft.
- David M. Houston conceived and designed the experiments, performed the experiments, authored or reviewed drafts of the article, and approved the final draft.
- Bernhard Lüthi conceived and designed the experiments, authored or reviewed drafts of the article, and approved the final draft.

- Philip J. Seddon conceived and designed the experiments, authored or reviewed drafts of the article, and approved the final draft.

## Field Study Permissions

The following information was supplied relating to field study approvals (i.e., approving body and any reference numbers):

This study complies with the relevant national, international, and institutional guidelines regarding animal care. It was conducted under a research permit (38882-RES) issued by the Department of Conservation under the New Zealand Wildlife Act 1953. All manipulations were approved by the Ethics Committee of the University of Otago (D69/17).

## Data Availability

The satellite data are available at the Movebank Animal Tracking: 691598094.

https://www.movebank.org/cms/webapp?gwt_fragment=page=studies,path=study 691598094.

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
