# Peer review of "A Jack of All Trades—Tawaki/Fiordland penguins are able to utilise diverse marine habitats during winter migration"

_PeerJ, doi:10.7717/peerj.19695_

## Round 0.1 · original submission · Major Revisions

Thank you very much for your manuscript titled “A Jack of All Trades - Fiordland penguins/Tawaki are able to utilise diverse marine habitats during winter migration” that you sent to PeerJ.

This study presents very valuable and relevant information on the winter migration patterns and non-breeding dispersal of Tawaki/Fiordland penguins from satellite telemetry monitoring, where it was observed that the penguins utilize a variety of oceanic habitats in contrast to the general dispersal patterns of other crested penguins.As you will see below, comments from referee 1 suggest a minor revision while reviewers 2 and 3 suggest a major revision before your paper can be published. Given this, I would like to see a major revision dealing with the comments. Their comments should provide a clear idea for you to review, hopefully improving the clarity and rigor of the presentation of your work. I will be happy to accept your article pending further revisions, detailed by the referees.

Reviewer 1 suggests checking the English and spelling of the writing, in addition to clarifying some questions that the reviewer has about the interpretation of the observations..

Reviewer 2 suggests improving the language of the writing, improving the editing of figures, tables, discussion, and conclusions. He also has observations on the study's perspective.

Reviewer 3 is generally satisfied with the manuscript, however, they point out several points that require further development to strengthen the interpretation and transparency of the modeling approach.

Please note that we consider these revisions to be important and your revised manuscript will likely need to be revised again.

·

Basic reporting

Check the English in the Abstract, it may have been written in a rush as there are a few grammatical errors (“a common…” and “breed of southern New Zealand”), the rest of the text seems fine.
But be consistent with the spelling of names such as Subantarctic Front (mostly in capital letters, some in minor case (eg L 268).
Also be consistent in the use of location names, Milford Sound, Fjordland and Harrison Cove are all used for the same location and readers not familiar with the site might get confused, rather stick to one name after introducing the site (with its more general location like Fjordland & Milford Sound). For example, Fig 2 uses all three names in the legend and graph.

Experimental design

no comment

Validity of the findings

Maybe discuss how tawaki can leave 2 months earlier than Snares penguins to have more time to reach the PFZ? What are the differences in breeding phenology?

Regarding birds from different sites reaching more or less the same areas, is there anything known about breeding site fidelity or movement between breeding sites by individual birds?

Additional comments

Maybe not important to mention in the manuscript, but any suggestions why the two birds from Oamaru did not transmit for longer? Do you expect them to have died? Have they been resighted? Could an early mortality be linked to them having been in rehabilitation (if that was the case). Or could it be linked to the location of Oamaru in comparison with the other breeding sites (east coast vs west coast)?

·

Basic reporting

Well-written but language could be improved in some points. The word dispersal is misplaced and should be replaced with migration.
Good referencing but very much penguin-focused.
Professional structure of article, but figures and tables lack captions, and end of discussion may provide a view of the results in a broader context as well (very much penguin-focused).

Experimental design

Fits within the scope of the journal.
Knowledge gap not stated very clearly, as previous studies have already found wintering areas (but less precise) and the important factors determining habitat choice.
Last research question is a bit strange, because in order to assess an advantage, one needs to know what the outcome of alternatives would be, and these are not tested. It remains speculative.
Investigation is mostly rigorous, but it is a shame that the analysis could only be done on a year-level. Which insights may you miss because of variables on year-level? Some graphs are not suited for the purposes as used in the text (see comments on fig. 2).
Methods well-described.

Validity of the findings

Sample sizes were sometimes low in a group. Replication should be encouraged.
I have not seen the underlying raw data, but data are mostly described in a good way.
Conclusions are not really conclusions, instead they are mostly speculations following up on the research.

Additional comments

Please see my detailed comments in the attached file.
Very interesting work which would be a valuable contribution to the literature.

Reviewer 3 ·

Basic reporting

This is a very well-written and thorough manuscript. The language is clear, professional, and unambiguous throughout. The structure is logical, and the manuscript conforms to PeerJ standards.
The introduction provides strong context, and the background is appropriate and well-cited. Figures and tables are relevant and well-presented, though there are a few suggestions for improvement (see below). Raw data appear in MoveBank.
Minor Suggestions:
• Figure 1: Consider including all known breeding colonies, not just tagging sites, to improve spatial context.
• Table 1: Add a summary row with mean (± SD) across all individuals to aid in comparative interpretation.
• Line 129: Change “individual deployed” to “individually deployed”.
• Line 132: Consider rewording for clarity: “It was intended to deploy five devices at each of three sites…”
• Line 163: When listing R packages, please include package versions for reproducibility.
• Line 164: If a swimming speed value was used in any analysis, please report it explicitly.
• Line 240: Report the percentage of locations retained after filtering to give readers a sense of data reduction.
• Line 277: After "surface current velocity," the response variable is listed as SLA: 0.6%. This appears to be a labeling error and should likely read VELO: 0.6%.

Experimental design

This is original primary research, well within the scope of PeerJ. The research question is clearly defined and addresses an important gap in our understanding of Eudyptes pachyrhynchus non-breeding ecology. The methods are generally appropriate and described in good detail, with a few areas that could be clarified or extended.
Suggestions:
1. Foraging Behavior During Migration:
Given the long trip durations and movement variability, consider whether penguins may be foraging en route. Movement-based proxies (e.g., residence time, speed, turning angle, Hidden Markov Models) could help identify area-restricted search behavior. This would clarify which legs of the migration serve a foraging function or if birds are simply commuting to distant foraging grounds.
2. Threshold for Arrival at Destination:
Justify why distance rather than time threshold used to define “arrival” at a destination. This may better reflect behavioral transitions in migratory or foraging context.
3. Comparison with Pre-Moult Trips:
The pre-moult trips mentioned are unusually long. A quantitative comparison of pre- and post-moult trip characteristics (e.g., distance, duration, habitat, direction) would offer valuable insight into the extent and nature of migratory flexibility in this species.

Validity of the findings

The findings are novel, ecologically relevant, and well supported by the data. The conclusions are appropriately linked to the results. However, several areas need further development to strengthen interpretation and transparency of the modeling approach.

1. Model Evaluation Metrics:
The manuscript does not currently report standard metrics of model performance. Please at least include AUC (Area Under the Curve) to assess model discrimination and some metric of false positive and false negative rates to evaluate prediction accuracy and reliability.These metrics are standard practice for Maxent modeling and are essential for assessing robustness. See Ballard et al. (2012) for an applied polar example, and Elith et al. (2006, 2011) for Maxent-specific evaluation guidance.
Citations:
Ballard, G., Jongsomjit, D., Veloz, S.D., & Ainley, D.G. (2012). Coexistence of mesopredators in an intact polar ocean ecosystem: The basis for defining a Ross Sea marine protected area. Biological Conservation, 156, 72–82.
Elith, J., Graham, C.H., et al. (2006). Novel methods improve prediction of species’ distributions from occurrence data. Ecography, 29, 129–151.
Elith, J., Phillips, S.J., et al. (2011). A statistical explanation of MaxEnt for ecologists. Diversity and Distributions, 17, 43–57.

2. Dynamic Oceanographic Features and Mixed Layer Depth (MLD):
Many of the environmental variables used in the habitat model were incorporated as seasonal means, effectively treating them as static predictors. In reality, these variables—particularly MLD—can vary substantially at temporal scales relevant to penguin movements and foraging trips. For example, Ballard et al. (2019) demonstrated that variability in thermocline strength at the scale of individual dives directly influenced Adélie penguin foraging success. Given that MLD emerged as the strongest predictor in this study but was only available as a long-term composite, it would strengthen the interpretation if the authors discussed how within-season variability in MLD might affect their conclusions. Several recent studies (e.g. Kai et al, Veatch et al) on Lagrangian coherent structure highlight the impact that dynamic features can have on predator distributions.

Additionally, several penguin tracks show a consistent diagonal boundary at their distal ends, suggesting a potential constraint or avoidance zone. This boundary may correspond to a dynamic oceanographic feature, such as the polar front, whose position shifts seasonally and may not be adequately captured by the mean frontal positions depicted in Figure 1. If dynamic frontal data (e.g., derived from satellite altimetry or SST gradients) are available, it would be useful to compare these to the penguins’ observed tracks. Otherwise, briefly acknowledging this possibility would enhance the ecological interpretation of movement constraints.
Citations:
Ballard, G., Schmidt, A. E., Toniolo, V., Veloz, S., Jongsomjit, D., Arrigo, K. R., & Ainley, D. G. (2019). Fine-scale oceanographic features characterizing successful Adélie penguin foraging in the SW Ross Sea. Marine Ecology Progress Series, 608, 263–277.
Kai, E. T., et al. (2009). Top marine predators track Lagrangian coherent structures. PNAS, 106(20), 8245–8250.
Veatch, J. M., et al. (2025). Lagrangian coherent structures influence the spatial structure of marine food webs. Communications Earth & Environment, 6(1), 127.

3. Unexplored Suitable Habitat East of NZ:
The Maxent model predicts suitable habitat east of New Zealand, yet no penguins were tracked using that area. Please discuss possible reasons — overprediction, unmodeled ecological constraints (e.g., currents), or behavioral conservatism — and how this may affect the interpretation of model predictions vs. realized distributions.

---

## Round 0.2 · accepted · Accept

Thank you for submitting your manuscript entitled "A Jack of All Trades – Tawaki/Fiordland penguins are able to utilise diverse marine habitats during winter migration" to PeerJ.

After reviewing this revised version of your manuscript, I see that the main comments suggested by the reviewers have been included, while the suggestions not considered are justified in detail. Therefore, I am satisfied with the current version and consider it ready for publication.

·

Basic reporting

well written and previously noted language and naming issues resolved

Experimental design

no comment

Validity of the findings

no comment

Additional comments

The manuscript has been improved and all my previous concerns have been dealt with. I recommend the publication of this article.